# Uncovering transcriptional dark matter via gene annotation independent single-cell RNA sequencing analysis

Michael F. Z. Wang[1], Madhav Mantri [1], Shao-Pei Chou[2], Gaetano J. Scuderi[1], David W. McKellar[1], Jonathan T. Butcher [1], Charles G. Danko [2] & Iwijn De Vlaminck [1✉]

Conventional scRNA-seq expression analyses rely on the availability of a high quality genome annotation. Yet, as we show here with scRNA-seq experiments and analyses spanning human, mouse, chicken, mole rat, lemur and sea urchin, genome annotations are often incomplete, in particular for organisms that are not routinely studied. To overcome this hurdle, we created a scRNA-seq analysis routine that recovers biologically relevant transcriptional activity beyond the scope of the best available genome annotation by performing scRNA-seq analysis on any region in the genome for which transcriptional products are detected. Our tool generates a single-cell expression matrix for all transcriptionally active regions (TARs), performs single-cell TAR expression analysis to identify biologically significant TARs, and then annotates TARs using gene homology analysis. This procedure uses single-cell expression analyses as a filter to direct annotation efforts to biologically significant transcripts and thereby uncovers biology to which scRNA-seq would otherwise be in the dark.

[1] Nancy E. and Peter C. Meinig School of Biomedical Engineering, Cornell University, Ithaca, NY 14853, USA. [2] Baker Institute for Animal Health, College of Veterinary Medicine, Cornell University, Ithaca, NY 14853, USA. ✉email: vlaminck@cornell.edu

Single-cell RNA sequencing (scRNA-seq) is a powerful tool to study development, tissue, and disease biology[1–9]. scRNA-seq analyses currently rely on the availability of high-quality genome annotations to define cell features and to perform cell clustering, dimensionality reduction, differential gene expression, and other analyses[10–14]. Identifying genes and correctly annotating their locations and coding regions within the genome is a difficult task that collaborative projects such as Ensembl[15], ENCODE[16], and GENCODE[17] are striving to achieve. Gene annotations are generated using homology analyses, updated based on experimental evidence, and manually curated for select organisms such as human, mouse, and zebrafish[15]. For less routinely studied organisms, fewer RNA-seq datasets are available and gene annotations are dependent on purely computational approaches such as phylogenetic sequence comparison with minimal verification. Consequently, as we show here with scRNA-seq experiments and analyses spanning six species (human, mouse, chicken, gray mouse lemur, naked mole rat, and sea urchin), significant number of genes and cell-type-specific information is missed in current scRNA-seq workflows.

To overcome this hurdle, we propose a procedure to perform scRNA-seq data analysis in the absence of a high-quality gene annotation (Fig. 1a). This procedure first identifies transcriptionally active regions (TARs) in scRNA-seq data using groHMM[18]. Briefly, groHMM uses a hidden Markov model to divide the genome into two states (transcribed and untranscribed) based on read coverage in each bin. These HMM-derived TARs capture the transcriptional activity in the sample, limited only by the scRNA-seq measurement and quality of the genome assembly. Single-cell TAR expression analyses are then performed, using both annotated TARs (aTARs) and unannotated TARs (uTARs), revealing the degree of biologically relevant information that exists both inside and outside of known gene annotations. As expected, we find that the degree of relevant information in uTARs is strongly dependent on organism, tissue, cell type, and stage of development. uTARs that are differentially expressed across different cell clusters are retrieved and annotated based on nucleotide sequence homology ("Methods"). We employed this approach to uncover uTARs that distinguish canonical cell types during early stages of chicken embryonic heart development, and that define cell types in the gray mouse lemur, naked mole rat spleen, and sea urchin embryo.

While groHMM has previously been used to annotate regions of transcriptional activity from nascent transcription in *Arabidopsis*[19] and other species[20] that are produced in bulk, our approach annotates transcriptional activity that is cell-type-specific within scRNA-seq data. The approach described here can be applied in two ways. First, our procedure enables more comprehensive single-cell analysis when a high-quality gene annotation is not available. Second, our results point to an application for scRNA-seq assays to create, curate, and correct gene annotations, where single-cell expression analysis is used as a filter to direct annotation efforts to those features which are biologically significant.

## Results

### Gene-annotation-independent scRNA-seq analysis.

We developed a gene-annotation-independent scRNA-seq analysis and benchmarked it using publicly available and new datasets generated on the 10X Genomics platform and a Smart-seq2-generated dataset ("Methods"). We explored different genome assemblies and annotations, including recent assemblies for human, mouse, chicken, gray mouse lemur, naked mole rat, and sea urchin, as well as prior annotations for human (Fig. 1c, d). The size of the genome assemblies ranged from ~1 billion base pairs to ~3 billion base pairs (Fig. 1b). As a proxy for the completeness of different gene annotations, we compared the number of annotated transcripts relative to the size of the genome assembly (Fig. 1c). We found that less routinely studied organisms such as the naked mole rat and sea urchin, as well as an older human genome annotation (RefSeq hg16 annotations for the hg16 reference assembly) have a lower number of annotated transcripts relative to the genome assembly size, as expected. We next quantified the number of sequences that align outside the gene annotation (assigned to uTARs, see below). For a recent human gene annotation (GENCODE v30 hg38), only 2% of reads aligned outside of the annotation. Considerably less effort has been devoted to the genome annotations of less routinely studied species, as is evident from the percentage of sequences that fall outside of the most current genome assemblies for gray mouse lemur (7.1%), naked mole rat (4.2%), and purple sea urchin (10.7%) (Fig. 1d). We examined older human genome assembly releases and found a decrease in the percentage of reads that mapped outside of annotations, as annotations improved from 2004 to 2019 (Fig. 1e).

We sought to perform scRNA-seq analysis based on all transcriptional products detected by the assay, including those that are not included in annotations. To identify TARs, we implemented a HMM after alignment of the RNA-seq data to the reference genome ("Methods")[18]. Figure 1f shows an example of TARs identified for human genome build 16 along a segment of chromosome 22 comprising *IGLL5*. Through comparison to the genome annotation, we labeled TARs as either annotated (aTARs) or unannotated (uTARs). Finally, we counted the number of transcripts for each TAR in each cell in a dataset, creating a digital expression matrix.

We used Seurat to analyze single-cell gene and TAR expression profiles[21,22]. We performed dimensionality reduction and single-cell clustering analysis on both TAR and gene expression profiles to determine the extent of biologically significant information contained within uTARs. We analyzed the Tabula Muris[7] 10X-generated dataset for the mouse spleen and kidney, human PBMC data available through 10X Genomics[23], a gray mouse lemur lung dataset, a publicly available naked mole rat spleen dataset[24], and a publicly available sea urchin dataset[25]. In addition, we performed single-cell RNA-seq profiling of embryonic chicken heart tissue at different stages of development (days 4, 7, 10, and 14 post fertilization). Following data pre-processing ("Methods"), we implemented dimensionality reduction (principal component analysis, PCA) and used UMAP[26] to visualize cells in 2D. We labeled cell groups based on existing metadata or expression of canonical marker genes ("Methods"). We found a significant number of mm10 assembly bases annotated by uTAR features (Supplementary Fig. 1a). Comparison of UMAP visualizations of single-cell gene and uTAR expression profiles revealed that significant cell-type-dependent structure is retained in the uTAR UMAPs (Fig. 2a and Supplementary Fig. 1b). We computed silhouette values as a measure of the consistency within clusters of the data and found close agreement between silhouette values calculated for aTAR and gene-expression-based clustering, as expected, and also good agreement between uTAR and gene-expression-based clustering for several datasets (Fig. 2b). This observation suggests that scRNA-seq reads outside of gene annotations contain significant biological information that can accurately separate cell types. Dimensional reduction on human PBMC uTAR expression reveals significant structure for the older hg16 genome build, but not for the most recent hg38 build, as expected, given that hg16 is older and has less comprehensive annotation (Supplementary Fig. 1b). Dimensional reduction on mouse spleen uTAR expression reveals structure in many cell types such as T and B cells while dimensional reduction on mouse

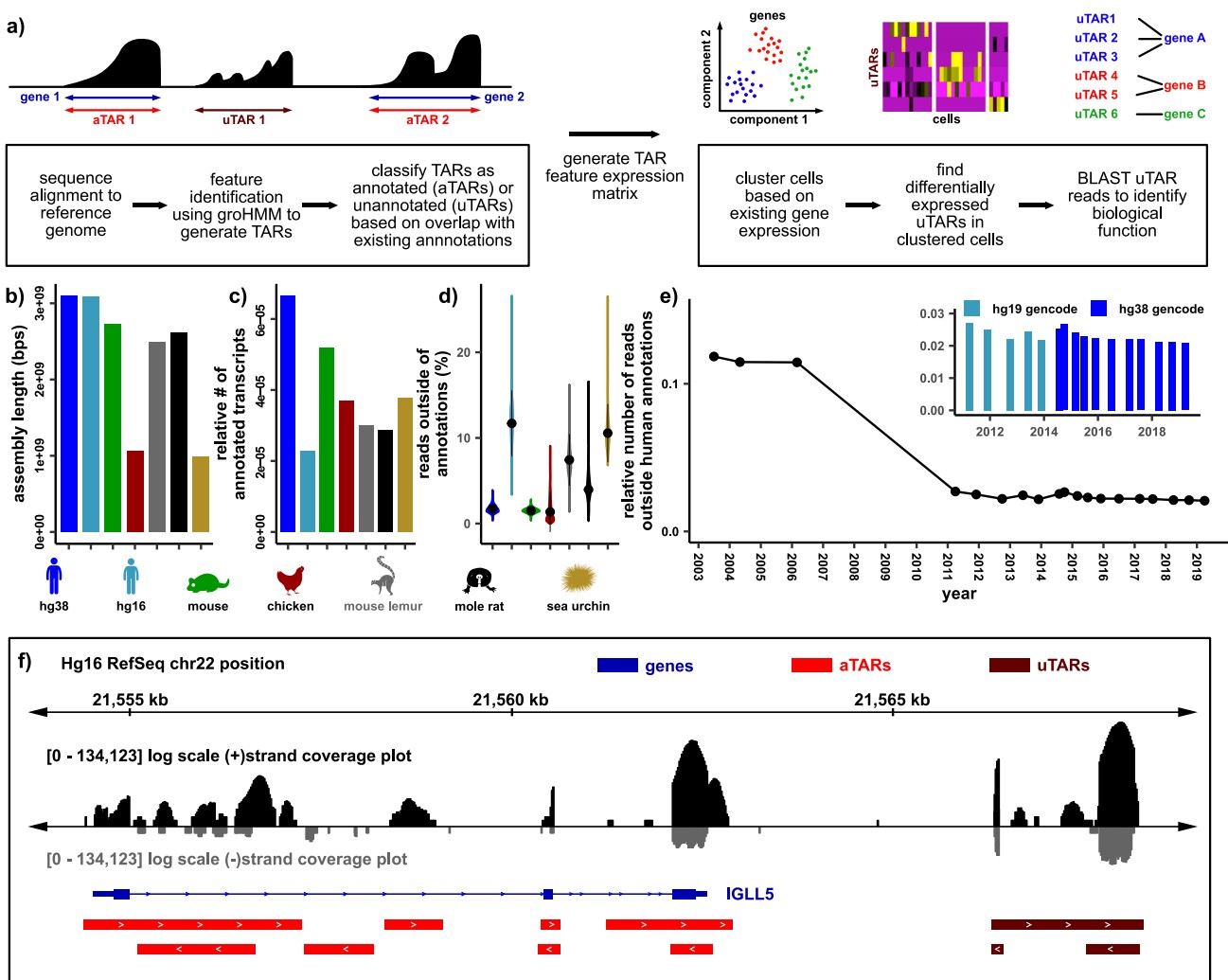

**Fig. 1 Generating de novo features based on genome coverage. a** Workflow to generate TARs and to identify biologically meaningful uTARs. **b** Total genome assembly sequence length for human (hg38 and hg16), mouse (mm10), chicken (GRCg6a), gray mouse lemur (Mmur_3.0), naked mole rat (HetGla_1.0), and sea urchin (Spur_4.2). **c** Total number of annotated transcripts in existing annotations normalized to the assembly sequence length for humans (hg38 GENCODE v30, hg16 RefSeq), mouse (GENCODE vM21), chicken (GRCg6a Ensembl v96), gray mouse lemur (Mmur_3.0 RefSeq), naked mole rat (HetGla_1.0 RefSeq), and sea urchin (Spur_4.2 RefSeq). **d** Relative number of unique scRNA-seq reads outside of gene annotations contained in uTARs for each cell shown as violin plots (3849 cells in hg38 and hg19, 6113 in mouse, 14008 in chicken, 6321 in lemur, 2657 in naked mole rat, 2658 in sea urchin). Mean values (black dots) and 2 standard deviations above and below the mean (black bars) are shown. **e** Relative number of unique scRNA-seq reads outside of gene annotations for different human genome assemblies and annotations at different times (3849 cells). **f** Example of groHMM defined aTAR (red) and uTAR (maroon) features along hg16 chr22 with RefSeq hg16 gene annotations shown in blue. Sense strand coverage plotted in black while antisense strand coverage plotted in gray (log-e scale).

kidney uTARs reveals less structure. For gray mouse lemur lung, naked mole rat spleen, and sea urchin embryo, we find that dimensional reduction on single-cell uTAR expression profiles reveals near-identical structures to that of annotated gene expression profiles. We conclude that for species that are not routinely studied and for which high-quality gene annotations are not available, extensive biologically significant information is missed by conventional scRNA-seq analysis. For chicken heart tissues, we find that cell-type clustering is conserved more strongly in uTARs for tissues collected at earlier stages in development. Day 4 uTARs perform the best in terms of separating cell types relative to other days while day 14 uTARs perform the worst. This observation reveals a high prevalence of unannotated transcripts in early progenitor cell types while the transcriptional programs of mature cell types in day 14 are captured much better by the current best annotation. This

suggests that the current chicken gene annotations do not define the complete transcriptional state of early embryonic tissue where many progenitor and transitional cell types are present. The extent of cell-type-specific information that is missing from standard genome annotations depends on the genome annotation, organism, tissue type, and stage of development.

We next evaluated the agreement between the most significant uTARs identified based on the total expression level from pseudo-bulk RNA-seq analysis and uTARs identified based on principal component (PC) loading values which correlate with their ability to resolve cell types in single-cell analysis. There is little overlap between uTARs with expression level greater than 10,000 reads (across all cells, pseudo-bulk analysis, "Methods") and uTARs with the highest loading values in the first 5 PCs calculated from scRNA-seq analysis (PCs, loading values greater than 0.5, 329 out of 3618 uTARs shared, Fig. 2c). In addition, we observed no

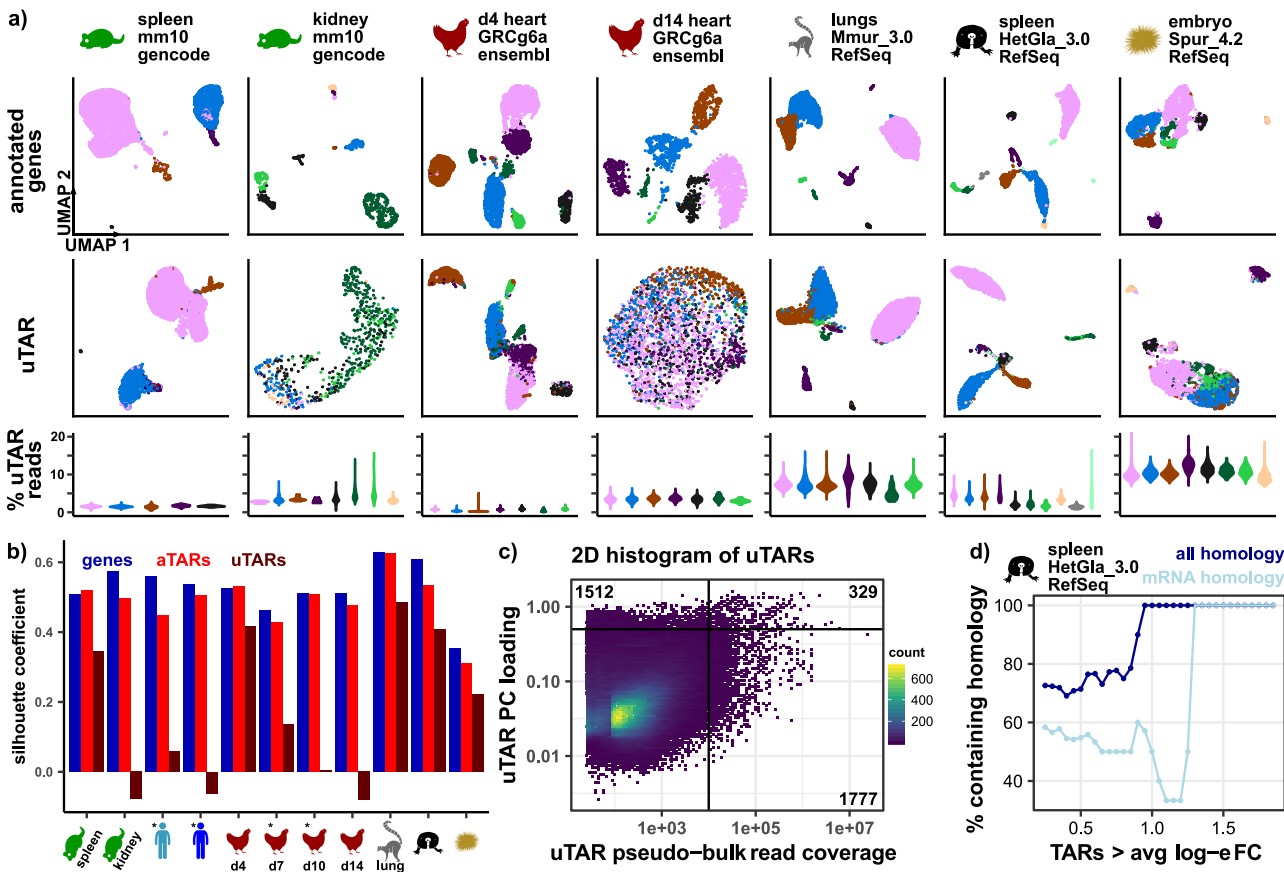

**Fig. 2 Reads in uTARs can separate cell types in different organisms. a** UMAP dimensional reduction on annotated gene expression features (top row) and uTARs (second row) for mouse spleen, mouse kidney, different time points in chicken embryonic heart development, gray mouse lemur lung tissue, and sea urchin embryonic tissue. Cells are colored in each column based on gene expression clustering. Relative number of uTAR reads for each cell in every cluster also shown as violin plots (third row, colors correspond to UMAPs); 6113 cells in mouse spleen, 610 cells in mouse kidney, 4365 in chicken day 4, 2198 in chicken day 14, 6321 in gray mouse lemur lungs, 2657 in naked mole rat spleen, and 2658 in sea urchin embryo. **b** Silhouette coefficient values based on 2D UMAP coordinates of gene expression (blue), aTARs (red), and uTARs (maroon) for 11 samples. UMAPs for samples labeled with (*) are shown in Supplementary Fig. 1b. Cell labels are defined by gene annotation clustering. **c** Correlation between top 5 PC loadings and pseudo-bulk read coverage of uTARs across 11 samples. Horizontal line at uTAR PC loading = 0.5, vertical line at uTAR pseudo-bulk read coverage = 1e + 4, $r^2 = 4.0$e-3. Quadrant numbers represent the number of uTARs in respective quadrant. **d** Relative percentage of uTARs containing homology to any sequence (blue) and mRNA sequences (light blue) as a function of log-e fold change expression for each cell type in naked mole rat spleen data. BLAST sequence homology results relative to nucleotide collection database thresholds: mean uTAR peak query length = 686 ± 731 bps, uTAR peak percent identity > 71%, e-value < 0.053, bit score > 52.8.

correlation between the total expression level of uTARs (pseudo-bulk read coverage) and their loading values in the first 5 PCs (Fig. 2c, correlation $r^2 = 4.0$e-3, p-value < 2.2e-16). This analysis indicates that uTARs with high total expression are not necessarily those that best define cell types, demonstrating the utility of scRNA-seq analysis as a selection filter before annotation and validation.

We next used the Wilcoxon rank-sum test to identify uTARs differentially expressed in the naked mole rat spleen dataset ("Methods"). We identified the largest peak within each differentially expressed uTAR and identified homologous sequences to the peak sequence using BLAST ("Methods"). This analysis revealed that uTARs that are more differentially expressed in any cell type (i.e., higher average log-e fold change expression relative to all other cell types) are also more likely to have sequence homology to transcripts contained within the NCBI nucleotide collection database, many of which have high homology to mRNA transcripts (Fig. 2d). We proceed to name uTARs with high differential expression in our datasets based on the BLAST results with the highest e-values and bit scores.

**uTARs recover unannotated transcripts.** We proceeded to extract and annotate uTAR features that define cell types. Cell clusters were first labeled by cell type, based on the expression of canonical gene markers or existing metadata. uTARs that were expressed in a minimum number of cells in each cluster (25% for mouse and chicken day 4, 50% for naked mole rat, lemur, and sea urchin) and at least 0.25-fold difference compared to the rest of the cells in log-e scale (0.25-fold change for chicken day 4, 0.50-fold change for naked mole rat and mouse, 0.75-fold change for lemur, 1-fold change for sea urchin) were identified. These differentially expressed uTARs were labeled based on nucleotide sequence homology using BLAST[27,28] on coverage peaks within the uTAR ("Methods").

We explored uTARs that were differentially expressed between immune cell types within the spleen from the Tabula Muris dataset (Fig. 3a). We uncovered an uTAR differentially expressed in natural killer cells containing reads with high sequence homology to *GTH1* which plays a role in protecting cells from reactive oxygen species[29]. We also found an uTAR containing homology to *PRPF8*, differentially expressed in macrophages,

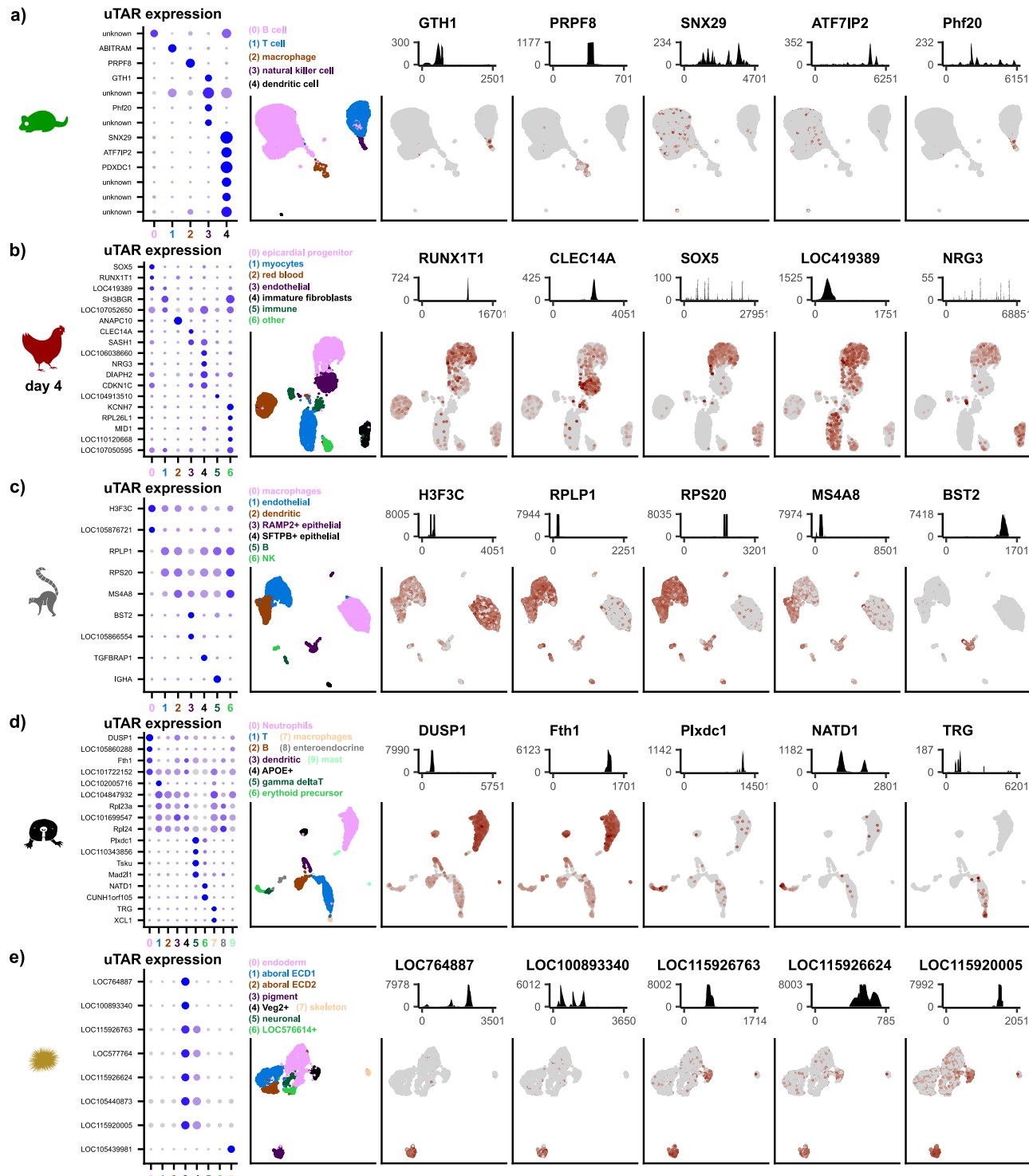

**Fig. 3 Biologically relevant information is contained in uTAR features.** Differential uTAR feature analysis for mouse spleen data (**a**), chicken heart day 4 data (**b**), gray mouse lemur *EPCAM*+ lung data (**c**), naked mole rat spleen data (**d**), and sea urchin embryo data (**e**). Dot plot (left) of differentially expressed uTAR features that are labeled based on sequence homology and cell clusters are numbered along the *x*-axis. Dot size corresponds to the percentage of cells that express the uTAR feature while darker blue color corresponds to higher level of log-e-normalized expression. UMAP (second left) colored and dimensionally reduced using gene expression features where cell clusters are labeled above the UMAP. Total coverage plot (top) of 5 uTARs along the length of the uTAR feature on the *x*-axis. The corresponding feature plot on UMAP projection is shown below the coverage plots where darker brown color correlates with higher log-e-normalized expression in each cell.

which is a pre-mRNA splicing factor that is essential for the catalytic step II in pre-mRNA splicing[30]. In addition, we found uTARs containing homology to *SNX29* and *ATF7IP2*, which were both upregulated in a small population of dendritic cells in the mouse spleen. *SNX29* is broadly associated with microtubule motor activity and phosphatidylinositol binding, and *ATF7IP2* is a transcription factor that couples other transcription factors to the transcription machinery. Both genes have predicted mouse orthologs but are unannotated in the GENCODE vM21 genome annotation.

We explored whether differentially expressed uTARs found in early stages of embryonic heart development were also differentially expressed in later stages. Silhouette coefficient analysis of the chicken dataset shows that uTARs better separate cell types in early development (Fig. 2a, b and Supplementary Fig. 1b). We identified 18 differentially expressed uTARs in the day 4 chicken heart with high differential expression and labeled them based on sequence homology (Fig. 3b, dot plot). These include *CLEC14A*, a potential tumor endothelial marker that plays a role in angiogenesis[31], *SOX5*, a transcription factor that regulates embryonic development and cell fate differentiation[32,33], and *RUNX1T1*, a transcriptional co-repressor that suppresses oncogenesis[34,35]. While *SOX5* and *RUNX1T1* are both annotated elsewhere in the chicken genome, we found uTARs with high sequence homology to these genes. The end of the *SOX5* uTAR is ~120 kb upstream of the *SOX5* annotation, which is ~340 kb in length, and the end of the *RUNX1T1* uTAR is ~50 kb upstream of the annotated counterpart, which is ~87 kb in length. This suggests that the *SOX5* and *RUNX1T1* uTARs may be additional exons of the existing annotations. When we examined the same set of differentially expressed uTARs in the day 14 chicken heart, we found that most were not differentially expressed (Supplementary Fig. 1c) suggesting a role for these differentially expressed uTARs in early development. Additionally, some differentially expressed uTARs in day 4 also had lower total read coverage in the day 14 dataset such as *RUNX1T1* and *CLEC14A* (Supplementary Fig. 1c, coverage tracks). Thus, uTAR analysis can identify potentially novel development-stage-specific transcripts.

We also explored differentially expressed uTARs in the gray mouse lemur lung, naked mole rat spleen, and sea urchin embryo datasets. Several differentially expressed uTARs were found in the gray mouse lemur *EPCAM+* lung cells including *H3F3C* and *BST2* (Fig. 3c). *H3F3C* is a novel unannotated gene that plays a role in maintaining nucleosome structure[36,37] while *BST2* is associated with interferon gamma and other cytokine signaling pathways in the immune system[38,39]. Differentially expressed naked mole rat uTARs were also identified within neutrophils, T cells, and macrophages (Fig. 3d). Homology analysis revealed that these included *DUSP1*, *NATD1*, and *TRG*. The *DUSP1* uTAR is a novel feature missing from the HetGla_1.0 with high total expression in neutrophils (5778 total unique scRNA-seq reads in 2657 cells). The protein encoded by *DUSP1* is a phosphatase with dual specificity for tyrosine and threonine and is a potential target for cancer therapy in humans[40,41]. We also found several uTARs that are highly expressed in the pigment cells of an embryonic sea urchin (Fig. 3e). They include several uncharacterized genes such as *LOC115926763* and *LOC115920005*. Therefore, our uTAR analysis reveals several unannotated features in understudied organisms such as the naked mole rat, gray mouse lemur, and sea urchin.

To test whether our analysis pipeline is compatible with full-length transcript data, we explored Smart-seq2 (SS2) full-length scRNA-seq data of gray mouse lemur testes tissue and detected several cell-type-specific uTARs ("Methods"). Dimensionality reduction on full-length uTAR features produces results in agreement with a gene-feature-based dimensional reduction,

similar to our observation for short-read sequencing scRNA-seq data (Supplementary Fig. 2a). We performed differential uTAR feature analysis for cell types within the testes tissue and found unannotated expression of *CCER1* in spermatocytes, *TBXAS1* in spermatogonium, and the ncRNA *LOC105867359* expressed in spermatids (Supplementary Fig. 2b).

The approach we report here can be used to uncover potentially novel cell subtypes by examining the differential expression of uTARs in canonical cell types. To test this concept, we examined a 10X dataset for the gray mouse lemur spleen tissue and found that uTAR features can be used to separate canonical cell types such as T cells, monocytes, and neutrophils, as expected (Supplementary Fig. 3a). Subsetting of plasma cells and clustering and dimensionality reduction based on uTAR expression revealed two distinct groups of plasma cells (Supplementary Fig. 3b). One of these two groups of cells, which was not identified by conventional analysis, was characterized by elevated expression of an uTAR with high sequence homology to the natural killer complex genes (*CD94*, *NKG2*, *Ly49L*) and uTARs with homology to *IGKV2* (Supplementary Fig. 3c) suggesting that these are *IGKV2+* plasma cells. This observation suggests that uTAR expression analyses can reveal cell subtypes that are not identified using conventional analyses.

Our approach conservatively labels TARs as uTARs when they are completely outside of gene annotations and label TARs as aTARs when they overlap with gene annotations but have opposite directionality. We expanded the definition of uTARs to include directionality and repeated the analysis of uTAR expression in developing chicken heart and observed similar behavior as described above (Supplementary Fig. 4). uTARs tended to cluster cell types better in early stages of embryonic development compared to later stages (Supplementary Fig. 4a, b). We identified 34 differentially expressed uTARs in the day 4 developing chicken heart where most lose their differential expression in later time points of development (Supplementary Fig. 4c). Of these uTARs, 11 overlap in position with an annotated gene but are transcribed from the antisense strand of a gene in the existing annotations. These features share a similar differential expression with the annotated counterpart in day 4 (Supplementary Fig. 4d) suggesting that the transcription of these antisense uTARs are correlated with the transcription of the corresponding sense gene features.

**Spatial transcriptomics reveals spatial co-expression of uTARs and canonical gene markers.** We used data generated on the 10X Genomics Visium spatial transcriptomics platform to visualize the spatial co-expression of differentially expressed uTARs with canonical gene markers. For this, 10 μm thick coronal tissue slices of embryonic heart at days 4 and 14 post fertilization were used for spatial transcriptomic analysis. The resulting data comprised 747 and 1995 barcoded spots for day 4 (5 hearts) and day 14 (1 heart), respectively. We found that the expression of a *SH3BGR*-related uTAR colocalized with that of the canonical cardiomyocyte marker *TNNT2* at both day 4 and day 14 (Fig. 4a, left), in line with observations from scRNA-seq (Fig. 3b and Supplementary Fig. 1c). We also observed clear expression of *RUNX1T1* uTAR in the day 4 tissue section but no expression in the day 14 section, in concordance with the scRNA-seq data that suggested stage-dependent expression of this uTAR (Fig. 4a, right). Expression of the *RUNX1T1* uTAR at day 4 colocalized with a subset of *COL1A1+* spots which is a marker for epicardial cells. Interestingly, the annotated *RUNX1T1* gene has almost no expression in the spatial data revealing a discrepancy between annotated gene expression and uTAR expression. We quantified the correlation between the spatial expression of 11 uTARs in the day 4 sample

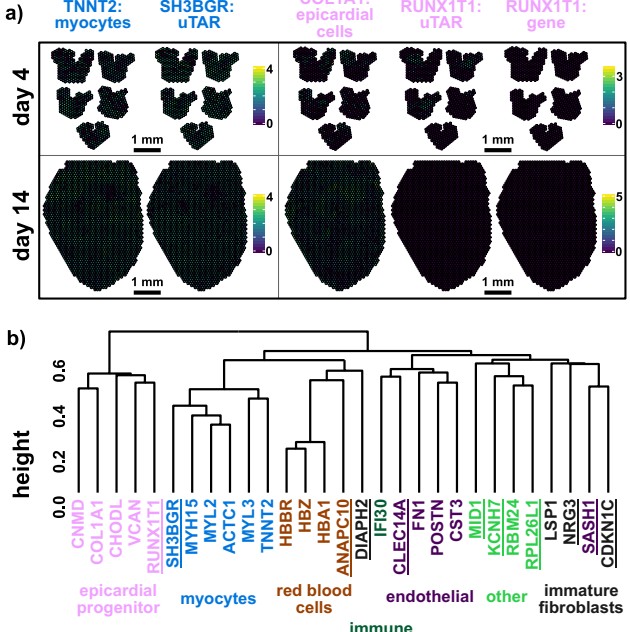

**Fig. 4 Spatial transcriptomics to map uTAR expression in chicken embryonic hearts. a** Spatial log-e-normalized expression of canonical *TNNT2* myocytes marker, *SH3BGR* uTAR, canonical *COL1A1* epicardial cells marker, *RUNX1T1* uTAR, and annotated *RUNX1T1* gene for chicken embryonic heart at day 4 (5 hearts) and day 14 (1 heart) post fertilization. **b** Dendrogram computed on Pearson correlation of log-e-normalized spatial expression for canonical gene markers and uTARs (underlined) in a day 4 chicken heart tissue section.

and several canonical gene markers such as *HBZ* for red blood cells, *TNNT2* for myocytes, and *COL1A1* for epicardial progenitors (Pearson correlation, log-e-normalized spot expression). We then performed hierarchical clustering on correlation values (Fig. 4b). We found that spatial uTAR expression in several cell types colocalized with canonical markers for those cells. Immune and immature fibroblast cell types are two of the three least abundant cell types according to the scRNA-seq data. As a result, their spatial correlation was low due to zero expression in most spots in the spatial data. Altogether, spatial transcriptomic data agreed with our findings of cell-type- and stage-specific expression of highly expressed uTARs.

## Discussion

Current scRNA-seq analysis relies on the availability of high-quality gene annotation to identify cell types, study dynamic transcriptional behavior over time, and predict gene interactions. We developed a method to identify uTARs that distinguish cell types in single-cell transcriptomic data. This approach is most relevant for the analysis of scRNA-seq datasets from less routinely studied organisms, and datasets from developing tissues that often comprise, as we show here, transcriptional products that are not represented in gene annotations. We present proof-of-principle applications of this procedure to study unannotated transcripts in embryonic development and to explore previously unknown genomic features in several species.

The strategy we propose broadly identifies TARs, without attempting to filter, curate, or characterize those regions, and then employs single-cell expression analyses to identify regions outside of the gene annotation that are relevant for the identification of cell states or types. Other TARs may encode transcripts that are broadly expressed in all cell types. This strategy

allows focusing annotation, curation, and validation efforts to those regions that are biologically significant. Our procedure expands the scope of scRNA-seq to tissues of understudied species, cell types, and development stages, enhancing the range of new biology that can be uncovered using these powerful approaches. In addition, our work points to an application for scRNA-seq to refine existing gene annotations, or to create de novo gene annotations.

## Methods

**Generation of chicken embryonic heart scRNA-seq data.** Fertile white leghorn chicken eggs were incubated using an egg incubator that controls humidity and temperature until the embryonic day of interest. Chicken embryonic ventricular free walls were excised aseptically in ice-cold Hank's balanced salt solution (HBSS) and then minced into 1 mm tissue fragments. Six dozen day 4, Hamburger-Hamilton developmental stage 21-23 (HH21-23) whole ventricles, four dozen day 7 (HH30-31) left and right ventricles, three dozen day 10 (HH35-36) left and right ventricles, and one dozen day 14 (HH40) left and right ventricles were, respectively, pooled for seven total samples to be analyzed via single-cell RNA sequencing. The day 4 and day 7 ventricular tissue fragments were digested under constant mild agitation at 37 °C in 1.5 mg/mL collagenase type II/dispase (Roche) for one cycle of 20 min and one cycle of 10 min, while the day 10 and day 14 ventricular tissue fragments were digested in 300 U/mL collagenase type II (Worthington Biochemical Corporation) for four cycles of 10 min to dissociate the cells from the tissue. Cells were then passed through a 40 μm filter and centrifuged into a pellet. For all samples, blood was removed by resuspension in an ACK lysis buffer followed by centrifugation and two washes in 1X phosphate buffered saline with 0.04% bovine serum albumin (1XPBS/BSA). For the day 14 samples, cells underwent fluorescence-activated cell sorting (FACs) to sort for live, non-apoptotic cells. Prior to FACs, the day 14 samples were first stained for 10 min in 4 μg/mL Calcein AM at 37 °C in HBSS with 2% fetal bovine serum (HBSS/FBS), washed in HBSS/FBS, centrifuged, stained with 1 μg/mL of 7AAD on ice in HBSS/FBS for at least 30 min, and then immediately cell sorted in HBSS/FBS. For all samples, single-cell suspensions were then resuspended at $1 \times 10^6$ cells/mL in 1XPBS/BSA prior to single-cell RNA sequencing. The 10X Genomics gene expression Library v2 Kit was used to isolate the day 10 and day 14 samples. The 10X Genomics gene expression Library v3 Kit was used to isolate 4DPF and 7DPF samples. Illumina NextSeq500 paired-end sequencing was used to sequence each sample. Chicken uTARs were generated by combining sequence alignment files for all samples. scRNA-seq analysis is shown for the right ventricle of day 7, 10, and 14.

**Sequence alignment and generation of expression matrices.** FASTQ sequence files were processed (adapter trimming, barcode tagging) using the Drop-seq[42] suite of computational tools and were aligned to respective genomes using STAR without gene annotation indexing. Gene expression matrices were generated based on gene annotations and TAR expression matrices were generated based on TAR annotations (described below) using Drop-seq tools.

**Identifying TARs from alignment files using groHMM.** We used groHMM[18] to predict transcribed regions from aligned scRNA-seq data in a strand-specific manner without annotations. Uniquely mapped reads from multiple scRNA-seq data were merged (i.e., without consideration for coverage of individual cells). We downsampled the combined alignment file of the Tabula Muris dataset and kept 5% of all uniquely aligned reads to account for the computational memory constraints of the groHMM tool. Then, we used groHMM to scan the mapped read counts along the genome with window size 50 bp without overlap. The emission probabilities were modeled by a gamma distribution. The gamma distribution parameters and transition probabilities were learned using the Baum-Welch expectation maximization (EM) algorithm. We predicted the transcribed regions from both the sense and antisense strands. Predicted transcribed regions within 500 bp were merged using bedtools[43] merge (parameters: -s -d 500). The coverage of each region was calculated using bedtools coverage (parameters: -s -counts -split). Transcribed regions with at least $n$ reads were kept and used as TAR features in the following analysis, where $n$ is set as 1/10,000,000 of uniquely mapped reads in the combined alignment file derived from SAMtools[44] view (parameters: -q 255).

TARs identified using the groHMM algorithm were labeled as annotated TAR (aTAR) or unannotated TAR (uTAR) features based on their overlap with existing gene annotations. The refFlat genome annotation format is used in all cases where the genomic start and end position of each transcript is recorded. HMM features that have any overlap with an existing gene annotation transcript based strictly on transcription start and stop sites were labeled as "aTAR", without accounting for strandedness. HMM features that overlap with an existing gene annotation but were on opposing strands were also labeled "aTAR". All other HMM features were labeled as "uTAR".

**Single-cell transcriptome analysis**. We used the Seurat suite of scRNA-seq tools[21,22] to analyze gene expression and TAR expression data. Cells were filtered based on the number of gene expression features, and TAR features were filtered based on being expressed in at least two cells. Cell filtering parameters were different for each dataset to optimize downstream analysis and to account for different protocols. Tabula Muris cells were filtered and scaled based on metadata information provided in the paper. Human PBMC cells were filtered based on having between 201 and 2599 gene features and less than 5% mitochondrial transcripts according to the hg38 assembly and GENCODEv30 annotations. Chicken cells were filtered based on having more than 200 gene features and less than 20% mitochondrial transcripts. Gray mouse lemur cells were filtered based on having between 201 and 2499 gene features. Naked mole rat cells were filtered based on having between 401 and 2499 gene features. Sea urchin cells were filtered based on having between 401 and 2499 gene features. Gene and TAR combined expression matrices were log-normalized and scaled before further processing. For the human PBMC dataset, 4000 variable features (including genes and TARs) were found based on mean–variance ratios. PCA was performed on variable gene features, aTARs, and uTARs in the human PBMC dataset. PCA was performed using all genes, aTARs, and uTARs in the other datasets. Cells were clustered using a KNN graph-based clustering approach with the first 10 PCs derived from gene expression reduction. UMAP projections were derived from the first 10 PCs using either gene expression, aTAR expression, or uTAR expression.

**Identifying differentially expressed uTARs**. We used the Wilcoxon rank-sum test to identify differentially expressed genes and TARs in every cell cluster, filtered based on increased expression as compared to all other cells. Canonical marker gene analysis was performed to identify cell clusters in days 4 and 14 chicken heart, naked mole rat spleen, and gray mouse lemur lung datasets. Differential mouse spleen uTARs were filtered by being in at least 25% of cells in the cluster and more than 0.5 log-e fold change compared to the rest of the cells. Differential day 4 chicken uTARs were further filtered by being in at least 25% of cells in the cluster and more than 0.25 log-e fold change compared to the rest of the cells. Lemur lung differential uTARs were further filtered based on being in at least 50% of cells in the cluster and more than 0.75 log-e fold change compared to the rest of the cells. Naked mole rat differential uTARs were further filtered based on being in at least 50% of cells in the cluster and more than 0.5 log-e fold change compared to the rest of the cells. Sea urchin differential uTARs were further filtered based on being in at least 50% of cells in the cluster and more than 1 log-e fold change compared to the rest of the cells. Lemur SS2 testes differential uTARs were filtered based on being in at least 25% of cells in the cluster and more than 0.25 log-e fold change compared to the rest of the cells (top 5 differentially expressed uTARs shown in Supplementary Fig. 2b, dot plot). Differential uTARs for the 10X lemur plasma cells from spleen were filtered based on being in at least 25% of cells in the cluster and more than 0.25 log-e fold change compared to the rest of the cells (4 uTARs shown in Supplementary Fig. 3c, dot plot).

**uTAR annotation using BLAST**. The total coverage of each differentially expressed uTAR is extracted using SAMtools depth and the coverage along the length of the uTAR is smoothed using locally weighted smoothing (LOESS) with the span set to 0.25. We then find the coordinates of the full-width at half-maximum (FWHM) of the largest peak along the length of the uTAR. The nucleotide FASTA sequence along this coordinate is extracted using samtools faidx. We then feed these sequences into BLASTn against the nucleotide collection (nt) database. BLASTn results are filtered for hits that have a maximum e-value of 0.01 and a minimum bit score of 50. Differentially expressed uTARs are then labeled based on hits with the highest bit score. uTARs that do not have hits that fit the e-value and bit score thresholds are labeled as unknown/undetermined. For the 10X lemur plasma cells, nucleotide FASTA sequences of uTARs were fed into IgBLAST against the human germline gene database. uTARs are similarly labeled based on hits with the highest bit score. For the SS2 lemur testes data, nucleotide FASTA sequences of full-length uTARs were fed into BLASTn against the nt database.

**Pseudo-bulk versus single-cell-derived uTAR comparison**. uTAR pseudo-bulk coverage was calculated using bedtools coverage on the merged alignment files for each experiment. uTAR PC loadings were calculated based on the sum of absolute PC loading values in the first 5 PCs based on uTAR PC analysis.

**10X Genomics Visium spatial transcriptome sample and library generation**. Whole hearts were dissected, placed in sterile HBSS, and perfused through the apex to remove blood. Fresh samples were embedded in Optimal Cutting Compound (OCT) and frozen with liquid-nitrogen-cooled isopentane before sectioning into 10 μm slices using Thermo Scientific™ CryoStar NX50 cryostat. Sections were mounted on −20 °C-cooled Visium slides. Five sections were mounted for day 4, four sections for day 7, two sections for day 10, and one section for day 14 post fertilization. cDNA libraries were generated using 10X Genomics Visium Spatial Gene Expression 3′ Library Construction V1 Kit. A 10X Genomics Visium Spatial Gene Expression slide has four capture areas. Each capture area has 5000 circular spots that are 55 μm in diameter with a center-to-center distance of 110 μm. Each spot contains DNA oligos consisting of a PCR handle, spot-specific spatial

barcodes, unique molecular identify (UMI), and poly-dT-VN tail for mRNA capture. Hematoxylin and eosin stained heart sections were imaged using Zeiss PALM MicroBeam laser capture microscope and images were processed using Fiji from ImageJ. Illumina NextSeq 500/550 was used to sequence the spatially tagged cDNA libraries with 150 cycle high-output kits (Read 1 = 120, Read 2 = 5, Index 1 = 14, and Index 2 = 8). Sequencing files were processed using 10X Genomics Space Ranger pipeline. TAR annotations were generated by running groHMM on the combined Space-Ranger-derived alignment files across the four time points. TAR and GRCg6a Ensembl v96 annotations were combined to create one set of genome annotation. Spatially tagged TAR and gene expression matrices were generated with the combined annotation set following the Space Ranger pipeline.

**Spatial transcriptome processing and visualization**. We used Seurat spatial transcriptome tools to visualize spatially tagged expression data. Feature expression matrices were log-e-normalized before further processing. We identified spatial uTARs that overlap with differentially expressed scRNA-seq uTARs in chromosome position and strandedness. If several spatial uTARs overlap with a scRNA-seq uTAR, we visualized and calculated spatial correlations based on the spatial uTAR with the highest total expression.

**Smart-seq2 analysis of gray mouse lemur testes data**. STAR aligned paired-end sequencing files were collected for each cell in the dataset. Read fragments from the paired-end sequencing data are collected using bedtools bamtobed with the "-bedpe" option set. We ran groHMM on read fragments to generate TARs. TARs were labeled as aTARs and uTARs in the same way as 10X-generated datasets. Labeled TAR annotation file in the gtf format is generated by a custom script provided in the TAR-scRNA-seq Git repository. The featureCounts[45] tool, with strandedness option set to 1 (-s 1) and paired-end option set (-p), was used to generate a count of each TAR features for every cell in the dataset. Count vectors were aggregated to generate a TAR expression matrix. We have included in our GitHub repository a script (scripts/SingleCellHMM_MW_SS2.bash) that will generate TAR annotations in refFlat and gtf formats from SS2 paired-end sequencing data.

**TAR-scRNA-seq tool and scripts**. The snakemake[46] pipeline along with the bash and R scripts used in the TAR-scRNA-seq tool is available at https://github.com/fw262/TAR-scRNA-seq with the identifier (https://doi.org/10.5281/zenodo.4567436).

**Reporting summary**. Further information on research design is available in the Nature Research Reporting Summary linked to this article.

# Data availability

Publicly available datasets: FASTQ files for the human PBMC dataset were downloaded directly from the 10X Genomics library of single-cell gene expression data. Tabula Muris alignment files (BAMs) for 10X Genomics droplet-generated data were downloaded from the Gene Expression Omnibus (GSE109774) and FASTQ files were extracted using the 10X Genomics bamtofastq tool[47]. Mouse uTARs were generated based on combining alignment files across all droplet-generated data and scRNA-seq analysis for kidney (10X_P4_5) and spleen (10X_P4_6) samples are shown. FASTQ files for the naked mole rat and sea urchin datasets were downloaded from GEO listed in their respective publications (GSM3885302 and GSE134350, respectively). Naked mole rat uTARs were generated based on combining alignment files for the spleen samples (SRR9291380, SRR9291381, SRR9291382, SRR9291383, SRR9291384, SRR9291385, SRR9291386, and SRR9291387) and scRNA-seq analysis for sample SRR9291380 (nmr_1.1) is shown. Sea urchin embryo uTARs were generate based on combining alignment files generated from SRR9693264, SRR9693265, and SRR9693266 and scRNA-seq analysis for sample SRR9693264 (D1) is shown. Gray mouse lemur lung tissue uTARs were generated from the Tabula Microcebus consortium by combining MLCA_ANTOINE_LUNG_EPCAM_POS_S12, MLCA_ANTOINE_LUNG_CD31_POS_S11, and MLCA_ANTOINE_LUNG_P3_S7 datasets, and scRNA-seq analysis for MLCA_ANTOINE_LUNG_EPCAM_POS_S12 is shown. Gray mouse lemur spleen tissue uTARs were generated from the Tabula Microcebus consortium from the MLCA_ANTOINE_SPLEEN dataset. Gray mouse lemur SS2 testes data were collected from the Tabula Microcebus consortium.

The chicken-related sequencing data discussed in this publication have been deposited in NCBI's Gene Expression Omnibus and are accessible through GEO Series accession number GSE149457. H&E stained tissue images for spatial RNA-seq datasets have been made available through the Git repository (https://github.com/fw262/TAR-scRNA-seq).

# Code availability

Code used to generate figures in this manuscript are available in the Git repository under the folder "scripts_for_figures." Git repository: https://github.com/fw262/TAR-scRNA-seq[48].

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

## Acknowledgements

We thank P. Schweitzer and colleagues at the Cornell Biotechnology Resource Center (BRC) for help with sequencing assays. We thank Phillip S. Burnham, Hao Shi, Alexandre P. Cheng, Adrienne Chang, Benjamin Grodner, Olga Botvinnik, Camille Sophie Ezran, Angela Wu, Mark Krasnow, and the Lemur Cell Atlas Consortium for discussions and feedback. This work was supported by R33CA235302 (to I.D.V.), R21AI133331 (to I. D.V.), DP2AI138242 (to I.D.V.), R01AI146165 (to I.D.V.), 1R01AI151059 (to I.D.V.), and a National Sciences and Engineering Research Council of Canada fellowship PGS-D2 (to M.F.Z.W.). The animal icons were created by Gan Khoon Lay (human), Vectorstall (mouse), Ines Simoes (chicken), Phạm Thanh Lộc (mouse lemur), Nova Hedberg (naked mole rat), and Vega Asensio (sea urchin) as part of the Noun Project[https://thenounproject.com/] under the CC BY 3.0 license[https://creativecommons.org/licenses/by/3.0/].

## Author contributions

M.F.Z.W., J.T.B., C.G.D., and I.D.V. designed the study. M.F.Z.W., M.M., and G.J.S. carried out the experiments. M.F.Z.W., M.M., S.C., and D.M. analyzed the data. M.F.Z.W. and I.D.V. wrote the manuscript. All authors provided feedback and comments.

## Competing interests

The authors declare no competing interests.
