## [Peer Review File · Nature Communications]

REVIEWER COMMENTS

Reviewer #1 (Remarks to the Author):

Dear authors,

your manuscript presents a workflow/pipeline for analysis of single cell RNA-Seq data. groHMM (by Danko et al.) is the central software component that allows finding transcribed regions within the new pipeline. Besides utilizing groHMM, the pipeline takes care of all the data pre- and postprocessing that is required to go from FASTQ-files to structural annotation of transcribed regions on a genomic level. In that way, the pipeline is unique and novel. The pipeline takes overlap of transcribed regions with an existing reference annotation into account if such an annotation is available for the expressed region. You demonstrate on several examples that the quality of reference annotations varies between species, and that information that can be gained from using the novel pipeline is larger for species with worse reference annotations (this is in my opinion to be expected).

The core idea of using groHMM for finding transcribed regions is not new. Kindgren et al. (2020, <https://academic.oup.com/nar/article/48/5/2332/5682908>) also employ groHMM to serve their purposes of finding TARs, they also check whether the TARs overlap with annotated genes or may represent novel transcribed regions. Anderson et al. (2020, <https://academic.oup.com/bioinformatics/article/36/9/2926/5698699>) even developed a transcript annotation package on the basis of groHMM. I am not saying that these two publications report the same as the presented new manuscript. But the core idea of using groHMM to find transcribed regions, and possibly checking whether they overlap with annotated genes, or even using them to annotate novel transcripts in the genome, is not new, anymore.

The pipeline itself is of novelty and of interest to others in the community where single cell RNA-Seq analysis is performed with non-model organisms because the pipeline is easy to use (if it runs reliably).

I am not completely sure that this work is material for a high impact journal such as Nature Communications because there is little novelty in the underlying methods. The presented workflow is designed in a smart way (and will probably work flawlessly after some more debugging) but it merely employs methods that were developed by others.

I see no flaws in the appropriateness and validity of statistical analysis. Results of this study can be reproduced with a minor effort by scientists who are experienced in using bioinformatics tools.

Other major concerns:

From my point of view, the most interesting finding of this manuscript is that a novel software pipeline for single cell RNA-Seq analysis exists. Many biologists who work with single cell sequencing data will be interested in running this pipeline. It is available at <https://github.com/fw262/TAR-scRNA-seq> and I have tested this software on 10 cores with the example data set from chicken. Regrettably, the software got stuck (over night) at "Finished job 11\\19 of 32 steps (59% done)". Since you claim that running the pipeline with this small data set should take about 15 minutes, something is wrong with the software. I strongly recommend that you test your software on a couple of more machines (maybe let colleagues who are not the developers test it) and debug... maybe it's about specific versions of the software dependencies. Checks for such things can be implemented. In any case, the software should be running reliably before publishing a paper about it.

From the abstract and the introduction, I had the impression that your pipeline will perform single-cell TAR expression analysis AND annotate some un-annotated TARs biologically using gene homology analysis (abstract) and gene prediction approaches (introduction). Looking at the methods part of your manuscript and at your git repository, I do not see that your workflow actually annotates the uTARs (other than structurally, i.e. the location is made clear). You say that BLASTn was performed to find sequence homology of uTARs (with e.g. mRNA entries in the

database). But this BLASTn step is to my understanding not part of the workflow (which does not list BLASTn as a dependency). I do not see gene prediction taking place anywhere in the paper. Maybe we define gene prediction differently. For me, gene prediction means that you define positions of a start codon, possibly of introns, and of a stop codon. This makes a predicted gene.

Minor issues:

It would be convenient if you provided the download links to the example data in your git repository instead of sending people to dig through long lists of possible file candidates.

Picard changes command line syntax, you should adapt your pipeline accordingly.

Reviewer #2 (Remarks to the Author):

Wang et al. extended the groHMM algorithm to perform annotation-free single cell expression analysis. The fact that the groHMM algorithm has been published weakens the novelty of the paper. However, the question that the authors were trying to address is very important in the single cell field. Solving such a question might provide potential novel insights in understanding fundamental biological questions. Meanwhile, the authors performed comprehensive and extensive benchmarking analyses, by surveying regular and spatial scRNA-Seq datasets profiled in multiple species and under diverse biological conditions, which I really appreciate and am very impressed. Thus, I would recommend the publication of the manuscript, after the authors address my following comments.

Major comments:

1. Seems the proposed approach can also be applied to bulk RNA-Seq analysis. It would be great if the authors can discuss about this to expand the scope of the paper.
2. The uTARs might convey biological information that is previously under-appreciated. Thus, I would suggest the authors to check whether novel cell populations can be discovered by the discovered uTARs. For instance, in Figure 2A panel Mmur, seems the purple population can be further separated by uTAR-only analysis. It would be great if some novel populations can be discovered, and some biological insights can be provided.
3. I would suggest the authors to analyze full-length sequencing datasets, besides 10X Genomics datasets, to further expand the scope of the paper to determine transcript isoforms of the detected novel genes.

Minor comments:

1. I would suggest the authors describe in detail about the procedure of 1) peak definition and selection, as well as 2) BLAST analysis, considering these procedures are critically important steps in their analytical pipeline.
2. Since the study depends heavily on the groHMM algorithm, I would suggest the authors give a comprehensive introduction, e.g. how reference genomes are binned, what are the states of the HMM model, how transition/emission probabilities are obtained?
3. Please include corresponding references for the Ensembl, ENCODE and GENCODE projects (line 32), as well as for the mouse lemur lung dataset (line 106).

Point-by-point address of the specific comments raised by the reviewers for manuscript ID: NCOMMS-20-28268

Our response in blue font. **Changes in the manuscript text are highlighted in yellow.**

Reviewer #1 (Remarks to the Author):

Your manuscript presents a workflow/pipeline for analysis of single cell RNA-Seq data. groHMM (by Danko et al.) is the central software component that allows finding transcribed regions within the new pipeline. Besides utilizing groHMM, the pipeline takes care of all the data pre- and postprocessing that is required to go from FASTQ-files to structural annotation of transcribed regions on a genomic level. In that way, the pipeline is unique and novel. The pipeline takes overlap of transcribed regions with an existing reference annotation into account if such an annotation is available for the expressed region. You demonstrate on several examples that the quality of reference annotations varies between species, and that information that can be gained from using the novel pipeline is larger for species with worse reference annotations (this is in my opinion to be expected).

The core idea of using groHMM for finding transcribed regions is not new. Kindgren et al. (2020, <https://academic.oup.com/nar/article/48/5/2332/5682908>) also employ groHMM to serve their purposes of finding TARs, they also check whether the TARs overlap with annotated genes or may represent novel transcribed regions. Anderson et al. (2020, <https://academic.oup.com/bioinformatics/article/36/9/2926/5698699>) even developed a transcript annotation package on the basis of groHMM. I am not saying that these two publications report the same as the presented new manuscript. But the core idea of using groHMM to find transcribed regions, and possibly checking whether they overlap with annotated genes, or even using them to annotate novel transcripts in the genome, is not new, anymore.

The pipeline itself is of novelty and of interest to others in the community where single cell RNA-Seq analysis is performed with non-model organisms because the pipeline is easy to use (if it runs reliably).

I am not completely sure that this work is material for a high impact journal such as Nature Communications because there is little novelty in the underlying methods. The presented workflow is designed in a smart way (and will probably work flawlessly after some more debugging) but it merely employs methods that were developed by others.

I see no flaws in the appropriateness and validity of statistical analysis. Results of this study can be reproduced with a minor effort by scientists who are experienced in using bioinformatics tools.

We thank the reviewer for their appreciation of our work, and for their careful review and valuable suggestions.

The reviewer is correct, other methods have previously been reported to identify and annotate transcripts from bulk transcriptomic data. We have implemented groHMM, an algorithm previously developed by us to define transcriptionally active regions (TARs). However, we like to stress that this is not the core idea of the current paper. The core idea of the current paper is to use TARs as input to a single-cell expression analysis to identify and annotate TARs that are expressed with cell-type specificity, which has not been reported previously. As we show in the paper, with experiments and analyses that span a wide range of different species and datasets, this is a straightforward but powerful concept that enables identifying and annotating TARs that are biologically significant, in a way that is not possible using conventional bulk transcriptomic analyses alone. The strength of the single-cell expression technique is in part derived from the fact that there is no direct relationship between the expression level of a TAR (selection criterion for bulk analyses) and its cell type specificity, as we show in Figure 2C. The single-cell analysis further enables identifying novel cell subtypes based on uTAR expression, as we show in a new analysis and new figure, inspired by a suggestion from reviewer 2. Last, our approach expands the range of cell-type specific biology that can be uncovered from scRNA-seq data beyond what was

possible previously, as supported by the numerous novel TARs we report, which is important most significantly for understudied/non-model species.

As a last point, we would like to stress that we report the first application of this type of analysis to spatial transcriptomic data, which allowed us to confirm the cell-type specificity of TARs and visualize the spatial expression of previously unannotated TARs.

Other major concerns:

From my point of view, the most interesting finding of this manuscript is that a novel software pipeline for single cell RNA-Seq analysis exists. Many biologists who work with single cell sequencing data will be interested in running this pipeline. It is available at <https://github.com/fw262/TAR-scRNA-seq> and I have tested this software on 10 cores with the example data set from chicken. Regrettably, the software got stuck (over night) at "Finished job 11\19 of 32 steps (59% done)". Since you claim that running the pipeline with this small data set should take about 15 minutes, something is wrong with the software. I strongly recommend that you test your software on a couple of more machines (maybe let colleagues who are not the developers test it) and debug... maybe it's about specific versions of the software dependencies. Checks for such things can be implemented. In any case, the software should be running reliably before publishing a paper about it.

We have modified the pipeline to improve its compatibility with different operating systems. Specific packages and dependencies are listed in the GitHub page (<https://github.com/fw262/TAR-scRNA-seq>). We have tested and verified that the test dataset analysis works on a local machine that runs ubuntu OS, 6 cores (12 threads) and 15Gb of memory. We were able to replicate the error described by the reviewer and fixed this bug. A ubiquitous wait statement in the previous pipeline waited for the completion of log files which only happened at the end of the same script. This bug was operating system dependent, and as a result we did not identify it previously. We thank the reviewer for pointing this out. In addition to fixing this bug, we have listed the installation of bedtools, BLAST, and all required R packages.

From the abstract and the introduction, I had the impression that your pipeline will perform single-cell TAR expression analysis AND annotate some un-annotated TARs biologically using gene homology analysis (abstract) and gene prediction approaches (introduction). Looking at the methods part of your manuscript and at your git repository, I do not see that your workflow actually annotates the uTARs (other than structurally, i.e. the location is made clear). You say that BLASTn was performed to find sequence homology of uTARs (with e.g. mRNA entries in the database). But this BLASTn step is to my understanding not part of the workflow (which does not list BLASTn as a dependency). I do not see gene prediction taking place anywhere in the paper. Maybe we define gene prediction differently. For me, gene prediction means that you define positions of a start codon, possibly of introns, and of a stop codon. This makes a predicted gene.

The primary function of our pipeline is to identify uTARs that are informative about specific cell-types and states. The pipeline uses BLASTn to aid in understanding the potential biological function of uTARs, especially in poorly annotated species. We agree that integrating these tools into our pipeline would be useful for users. We have therefore expanded the existing pipeline to perform scRNA-seq analysis using the Seurat suite of tools in R. Differentially expressed uTARs are first identified using a Wilcoxon Rank Sum test. The pipeline then runs BLASTn on differentially expressed uTARs (full length of uTAR) and uTAR peaks against the nt database. We then label uTARs based on the BLASTn result with the highest bit-score. These steps can be altered by users, to examine the effect of different clustering parameters, for example. The pipeline now returns a list of labeled differentially expressed uTARs and BLASTn results that are used to label those uTARs, in addition to the TAR and gene expression matrices. We have edited the introduction (second paragraph) to clarify that the pipeline annotates uTARs based on nucleotide sequence homology against the NCBI nt database rather than perform full gene prediction that includes start/stop codon and exon annotations, which is impractical from single-ended 5' short read sequencing data.

Minor issues:

It would be convenient if you provided the download links to the example data in your git repository instead of sending people to dig through long lists of possible file candidates.

The example dataset is provided in the “testData_small” folder in the git repository (<https://github.com/fw262/TAR-scRNA-seq>). It includes the fastq files for the day 4 and day 7 right ventricle chicken heart samples downsampled to 0.25% of the original dataset. The original dataset can be accessed on the Gene Expression Omnibus, accession number GSE149457. The link to the full dataset is also included in the github README page.

Picard changes command line syntax, you should adapt your pipeline accordingly.

We have changed the picard tools commands in the snakemake pipeline to reflect the new syntax. Please note that our pipeline makes use of the Drop-seq suite of computational tools that implements old picard command line syntax. These Drop-seq tool commands (i.e. function to generate expression matrices) will continue to show a note that picard command line syntax is changing.

Reviewer #2 (Remarks to the Author):

Wang et al. extended the groHMM algorithm to perform annotation-free single cell expression analysis. The fact that the groHMM algorithm has been published weakens the novelty of the paper. However, the question that the authors were trying to address is very important in the single cell field. Solving such a question might provide potential novel insights in understanding fundamental biological questions. Meanwhile, the authors performed comprehensive and extensive benchmarking analyses, by surveying regular and spatial scRNA-Seq datasets profiled in multiple species and under diverse biological conditions, which I really appreciate and am very impressed. Thus, I would recommend the publication of the manuscript, after the authors address my following comments.

We thank the reviewer for the appreciation of our work. As the reviewer points out, our study addresses an important challenge for the single cell field: conventional scRNA-seq analyses fundamentally rely on the availability of a comprehensive gene annotation, yet, as we demonstrate in our paper, gene annotations often are inaccurate or incomplete, in particular for developing tissues, or tissues from understudied organisms. We solve this issue by performing single-cell expression analysis on any region in the genome for which transcriptional products are detected, and use groHMM, an algorithm previously developed by us to identify those products. Our work will have impact in two ways: First, our method can be readily adopted by others to extend the range of biology that can be uncovered via scRNA-seq. Second, our work points to a new application for scRNA-seq to create, curate and correct gene annotations.

Major comments:

1. Seems the proposed approach can also be applied to bulk RNA-Seq analysis. It would be great if the authors can discuss about this to expand the scope of the paper.

Thank you for the question. We have in fact applied the approach to the aggregate (bulk) sequencing data for all datasets: our approach first identifies transcriptionally active regions (TARs) from the aggregate (bulk) sequencing data, without considering single-cell tag information. We then leverage the TARs as features for scRNA-seq analysis to identify previously unannotated TARs (uTARs) that have cell-type specific expression or that define cell (sub)types. Of interest, our paper includes a direct comparison of bulk and single-cell RNA-seq based feature selection (Figure 2C). We found that there is no correlation between uTARs that are highly expressed in bulk and uTARs that distinguish cell types, indicating that uTARs with high total expression are not necessarily those that best define cell types. This analysis demonstrates the utility of scRNA-seq analysis as a selection filter before annotation and validation.

2. The uTARs might convey biological information that is previously under-appreciated. Thus, I would suggest the authors to check whether novel cell populations can be discovered by the discovered uTARs. For instance, in Figure 2A panel Mmur, seems the purple population can be further separated by uTAR-only analysis. It would be great if some novel populations can be discovered, and some biological insights can be provided.

We thank the reviewer for the great suggestion. To explore this possibility, we examined a new dataset of mouse lemur spleen tissue measured by 10X. We found that unannotated TAR expression reveals a subset of plasma cells with strong expression of previously unannotated natural killer complex genes and *IGKV2-28* in this dataset. This analysis therefore identifies a novel population of plasma cells expressing the *IGKV2-28* immunoglobulin repertoire, that is not identified using conventional analyses, conveying biological information that is underappreciated as suggested by the reviewer. We include this example as a new supplementary figure (Extended data Fig. 3) in the paper.

3. I would suggest the authors to analyze full-length sequencing datasets, besides 10X Genomics datasets, to further expand the scope of the paper to determine transcript isoforms of the detected novel genes.

Thanks for another great suggestion. In the revised version of the manuscript, we include an analysis of a gray mouse lemur testes dataset measured by Smart-seq2 (SS2, full length RNA sequencing). We modified our pipeline to run groHMM on read fragments (from the start of read 1 to the end of read 2) instead of individual reads in the paired SS2 sequencing data generating non-overlapping TARs from full length sequencing data. We found that uTAR features of 708 cells from the testes tissue grouped cell types similar to gene feature reduction. We also labeled several differentially expressed uTAR such as *CCSER1* in spermatocytes and *TBXAS1* in spermatogonium. We have included this analysis in the new version of the paper, and included the data as a supplementary figure (Extended data Fig. 2).

Minor comments:

1. I would suggest the authors describe in detail about the procedure of 1) peak definition and selection, as well as 2) BLAST analysis, considering these procedures are critically important steps in their analytical pipeline.

We have updated the Methods section of the manuscript to include more detail on how uTAR peaks are identified and how the BLAST analysis is used to label differentially expressed uTARs (section, uTAR annotation using BLAST).

2. Since the study depends heavily on the groHMM algorithm, I would suggest the authors give a comprehensive introduction, e.g. how reference genomes are binned, what are the states of the HMM model, how transition/emission probabilities are obtained?

We added a more detailed description of groHMM in the introduction, including possible states, bin sizes, and how transition and emission probabilities are obtained. Please note that in our original manuscript submission, we included a detailed description of how we implemented the groHMM algorithm, including the bin size (50bps), different states (transcribed and not transcribed), emission (modeled by gamma distribution) and transition (learned from Baum-Welch EM algorithm) probabilities in the methods section entitled "Identifying TARs from alignment files using groHMM".

3. Please include corresponding references for the Ensembl, ENCODE and GENCODE projects (line 32), as well as for the mouse lemur lung dataset (line 106).

We have added references for the projects listed in line 32. The gray mouse lemur datasets mentioned in the paper are available from the mouse lemur cell atlas consortium which will publish their single-cell atlas of the gray mouse lemur shortly. We will include references to these datasets as soon as they become available.

REVIEWERS' COMMENTS

Reviewer #1 (Remarks to the Author):

Dear authors,

thank you for your efforts. You have addressed all issues appropriately from my point of view. I also confirm that your software now runs smoothly on my machine.

Thank you!

Reviewer #2 (Remarks to the Author):

I thank the authors for their responses! They have now addressed all my comments. I'm especially impressed by the identification of potential new cell types using uTARs discovered by their algorithm. I thus would recommend the publication of the manuscript.

One minor thing I want to clarify regarding my previous major comment #3. I realized that "determine transcript isoforms", even with full length scRNA-Seq approaches, might not be possible, considering 1) degradation might confound the genomic coordinates of transcript ends, e.g. as shown in the RNA velocity paper, and 2) splicing events could also happen at the middle of gene bodies, e.g. fruit fly *Sxl* gene. I apologize for my ignorance. Anyway I really appreciate that the authors analyzed full length scRNA-Seq data, which significantly improved the compatibility of their algorithm.